# Total Cholesterol and Mortality in Older Adults: A Sex-Stratified Cohort Study

**DOI:** 10.3390/nu17193128

**Published:** 2025-09-30

**Authors:** Maria Serena Iuorio, Diana Lelli, Stefania Bandinelli, Luigi Ferrucci, Claudio Pedone, Raffaele Antonelli Incalzi

**Affiliations:** 1Research Unit of Geriatrics, Fondazione Policlinico Universitario Campus Bio-Medico, Via Alvaro del Portillo, 200, 00128 Rome, Italy; mariaserena.iuorio@policlinicocampus.it (M.S.I.); c.pedone@policlinicocampus.it (C.P.); r.antonelli@policlinicocampus.it (R.A.I.); 2Residency Program in Geriatrics, Università Campus Bio-Medico di Roma, Via Alvaro del Portillo, 21, 00128 Rome, Italy; 3ASL Azienda Sanitaria Locale Toscana Centro, 50142 Florence, Italy; stefania1.bandinelli@uslcentro.toscana.it; 4Intramural Research Program, National Institute on Aging, National Institutes of Health, Baltimore, MD 21224, USA; ferruccilu@grc.nia.nih.gov; 5Research Unit of Geriatrics, Department of Medicine and Surgery, Università Campus Bio-Medico di Roma, Via Alvaro del Portillo, 21, 00128 Rome, Italy; 6Research Unit of Internal Medicine, Department of Medicine and Surgery, Università Campus Bio-Medico di Roma, Via Alvaro del Portillo, 21, 00128 Rome, Italy

**Keywords:** total cholesterol, mortality, older adults, sex differences, frailty

## Abstract

**Background**: The relationship between total cholesterol (TC) levels and mortality in older adults is complex and may differ from younger populations. While hypercholesterolemia is a known midlife risk factor, this association may weaken or reverse with age. Biological differences in cholesterol metabolism—particularly hormonal changes—may contribute to sex-specific mortality risks, but this remains underexplored. We examined the association between TC and all-cause mortality in older adults, assessing sex-specific differences. **Methods**: We used data from the InCHIANTI study, a longitudinal, population-based study conducted in Tuscany, Italy. From the original cohort (N = 1453), 999 participants ≥ 65 years with baseline TC and mortality data were included. TC levels were categorized as <200 mg/dL, 200–239 mg/dL, and ≥240 mg/dL. The primary outcome was all-cause mortality over 6-years. Kaplan–Meier curves and Cox proportional hazards models assessed mortality risk across TC categories in the overall population and by sex. Restricted cubic splines explored non-linear associations. Models were adjusted for age, sex (only in overall population), BMI, physical activity, diabetes, COPD, hypertension, eGFR, polypharmacy and frailty. **Results**: A threshold effect was observed: mortality risk rose sharply below ~200 mg/dL and remained stable above. Compared to the <200 mg/dL group, intermediate and high TC levels were associated with lower mortality risk (HR 0.72; 95% CI: 0.53–0.99 and HR 0.71; 95% CI: 0.49–1.02, respectively). In sex-stratified analyses, this pattern was pronounced in women but weaker and not statistically significant in men. Results held after excluding statin users and were confirmed by spline analysis. **Conclusions**: In older adults, particularly women, low TC may signal underlying vulnerability, including malnutrition or inflammation.

## 1. Background

The relationship between total cholesterol (TC) serum concentration and mortality has been extensively studied, and hypercholesterolemia is a well-established risk factor for cardiovascular and all-cause mortality in young and middle-aged adults [1,2,3]. Consequently, cholesterol-lowering strategies have been widely adopted to reduce cardiovascular morbidity and mortality in the general population [4,5]. However, the association between TC levels and mortality in older adults appears to be more complex and less well understood, with studies indicating this relationship may weaken or even reverse at advanced ages. In older adults, TC may reflect overall health status rather than cardiovascular risk. For example, a study of adults aged 65–98 years reported a two-fold increase in mortality risk for those in the lowest TC quartile compared to the highest, with authors suggesting that low TC levels are strongly linked to frailty and subclinical diseases [6]. Additionally, a recent study on centenarians reported that higher TC levels were associated with a greater likelihood of reaching 100 years. The authors proposed that cholesterol levels in very old age may serve as a marker of long-term metabolic stability and nutritional health [7].

The mechanisms underlying this apparent paradox may be related to cholesterol’s role in maintaining cellular membrane integrity, hormone synthesis, immune function, and its link with nutritional status. Low TC levels have in fact been consistently associated with malnutrition risk across various validated nutrition screening tools [8,9]. This evidence positions low TC as a potential marker of overall health decline, particularly in the context of frailty and undernutrition.

Importantly, the potential for sex-specific differences in the association between TC and mortality remains underexplored, although some evidence suggests that these differences may be significant. For instance, Yi et al. reported that low cholesterol was more strongly associated with increased mortality in women, whereas this association was weaker or absent in men, particularly in older age groups [10]. Biological differences in cholesterol metabolism between men and women, driven largely by hormonal influences, provide a rationale for examining these associations separately. In women, estrogen plays a crucial role in regulating cholesterol homeostasis, influencing hepatic cholesterol synthesis, catabolism, and bile acid production [11]. After menopause, reduced estrogen levels impair cholesterol clearance, often leading to increased TC levels with age [12,13]. Thus, low TC in postmenopausal women may indicate an abnormal metabolic state, reflecting conditions like malnutrition, systemic inflammation, or chronic illness. In contrast, cholesterol metabolism in men is less hormone-driven and more influenced by visceral adiposity and metabolic syndrome [11]. Therefore, in men, low TC may result from medication use (e.g., statins), lifestyle changes, or other metabolic adjustments that do not necessarily signal poor health. Furthermore, even when men have low cholesterol levels, their greater lean body mass and metabolic reserves may reduce the extent to which TC serves as a marker of malnutrition or frailty [14]. These differences suggest that TC levels may have distinct implications for mortality risk in men and women, reinforcing the need for sex-specific analyses to better characterize these associations. Given these considerations, we aimed to evaluate the association between TC levels and all-cause mortality in community-dwelling older adults and to investigate whether this relationship differs between older men and older women. Additionally, identifying potential sex differences in this association may help to refine risk assessments in older men and women.

## 2. Methods

### 2.1. Data Sources and Sample Selection

The study population consisted of participants from the InCHIANTI study, which is a prospective population-based study of older persons, designed by the Laboratory of Clinical Epidemiology of the Italian National Research Council of Aging (INRCA, Florence, Italy). A detailed description of the sampling procedure and data collection method has been published elsewhere [15].

Briefly, data collection began in September 1998 and was completed in March 2000. Older adults were randomly selected from the population registries of Greve in Chianti, located in a rural area and the Village of Antella in Bagno a Ripoli, located outside the urban area of Florence, in the Chianti area of Tuscany (Italy).

Participants underwent home interviews followed by clinic visits for medical examination, clinical tests, and physical function assessments. Trained interviewers administered structured questionnaires on dietary intakes, household composition, social networks, economical status, education, and general information on health and functional status.

The INRCA Ethics committee approved the InCHIANTI study protocol (protocol number 14/CE, date 28 February 2000) and all participants provided written informed consent, including permission to access medical records/administrative data and to analyze stored biological samples. For the present study, we included participants aged 65 years or older with measured TC serum concentration levels at baseline. We analyzed data from participants who were evaluated at baseline and at the 6-year follow-up (2004–2006).

### 2.2. Study Variables

Baseline data included demographic characteristics (i.e., age, sex, years of education), lifestyle (i.e., smoking habits, alcohol consumption, physical activity), clinical conditions, laboratory measures, functional status, frailty, and medication use. Operational definitions were as follows.

BMI was calculated as kg/m^2^. Physical activity was dichotomized as low (≤~4 h/week of moderate activity) vs. higher (>~4 h/week). Alcohol intake was expressed as grams/day.

Prevalent hypertension, diabetes, chronic obstructive pulmonary disease (COPD), thyroid disease, chronic liver disease, and prior non-fatal cardiovascular events were ascertained from the study examination, medication review, and physician/record documentation, and coded as binary variables (present/absent). Serum creatinine was measured at baseline; estimated glomerular filtration rate (eGFR) was calculated using a creatinine-based CKD-EPI equation (without race adjustment) and expressed as mL/min/1.73 m^2^.

Fasting serum total cholesterol (TC), LDL-C, and HDL-C were measured with enzymatic assays (Roche Diagnostics) and reported in mg/dL. High-sensitivity C-reactive protein (hs-CRP) was reported in mg/L (log-transformed in adjusted models). Albumin (g/dL) was derived from the reported albumin percentage multiplied by total protein (g/dL).

Functional status was assessed using Basic activities of daily living (ADL) and instrumental activities of daily living (IADL), and medication use [16,17]. We derived ‘ADL lost’ as the count of basic ADL tasks not performed independently (range 0–6) and ‘IADL lost’ as the count of instrumental ADL tasks not performed independently (range 0–8); higher scores indicate greater functional limitation. ADL tasks include bathing, dressing, toileting, transferring, continence, and feeding; IADL tasks include telephone use, shopping, food preparation, housekeeping, laundry, transportation, medication management, and finances.

Frailty was defined according to Fried’s frail phenotype as the presence of three or more of the following criteria: unintentional weight loss (10 lbs in past year), self-reported exhaustion, reduced grip strength, slow walking speed, and low physical activity [18].

Medication use was retrieved from records/interviews. Polypharmacy was defined as the concurrent chronic use of five or more medications [19].

### 2.3. Study Outcome

Primary outcome was all-cause mortality. Mortality data were collected using data from the Mortality General Registry of the Tuscany Region, and the death certificates from the Registry office of the municipality of residence.

### 2.4. Statistical Analysis

All analyses were conducted in overall population and then separately in men and women to account for potential sex-specific differences in the relationship between TC levels and all-cause mortality. TC serum levels were categorized into three groups (<200 mg/dL, 200–239 mg/dL, ≥240 mg/dL), using levels < 200 mg/dL as reference. These thresholds were chosen based on a combination of clinical relevance and consistency with both international guidelines and population-based data. The <200 mg/dL cutoff reflects the widely accepted definition of desirable cholesterol levels, while ≥240 mg/dL is traditionally considered high-risk for cardiovascular disease, according to established cardiovascular prevention and cholesterol blood management guidelines such as NCEP ATP III classification and National Health and Nutrition Examination Survey, while the intermediate range (200–239 mg/dL) corresponds to borderline-high values often used in clinical stratification [20,21,22]. Furthermore, these categories are aligned with the distribution of TC levels observed in our study population and are consistent with previously reported median values in older Italian adults, where average TC levels were approximately 225–230 mg/dL [23]. This combined rationale ensures both clinical applicability and population specificity for our analysis.

The risk for all-cause mortality across categories of TC levels was evaluated using Kaplan–Meier survival curves, which were compared with the log-rank test. A Cox proportional hazard model was used to estimate hazard ratios (HR) and 95% confidence intervals (CI) for all-cause mortality. This model was adjusted for potential confounders, including age, sex, physical activity, BMI, diabetes, chronic obstructive pulmonary disease (COPD), hypertension, estimated glomerular filtration rate (eGFR), frailty, and polypharmacy. Further models were performed adjusting also for serum albumin (g/dL), high-sensitivity C-reactive protein (hs-CRP; mg/L), thyroid and liver diseases, alcohol consumption (g/day) and statin use. These variables were selected based on their well-established associations with both cholesterol levels and all-cause mortality in older adults, as documented in previous epidemiological studies [24,25].

To mitigate potential reverse causation, we performed a 1-year lag sensitivity analysis: participants with follow-up ≤ 365 days were excluded and all Cox models were re-fitted in the overall cohort and separately in men and women.

Additional analyses were conducted to assess the impact of statin use by excluding participants on statin therapy.

Restricted cubic spline (RCS) regressions were employed to evaluate the dose–response relationships between continuous TC levels and hazard ratios for all-cause mortality, allowing for the visualization of potential non-linear associations. The primary specification used 3 knots placed at the 10th, 50th, and 90th percentiles of the TC distribution; hazard ratios (HRs) were centered at the reference TC value (HR = 1 at the reference). Non-linearity was assessed with a Wald test on the spline terms. As robustness checks, we also re-estimated the RCS models using Harrell’s 5-knot scheme (knots at the 5th, 27.5th, 50th, 72.5th, and 95th percentiles of TC).

All adjusted models used complete-case analysis. All tests were two-sided with α = 0.05. We report point estimates with 95% confidence intervals (CIs). Results with *p* < 0.05 were considered statistically significant. Finally, we assessed the association between TC and cardiovascular mortality, defined as death for ischemic stroke, or cardiovascular disease, using a Cox proportional hazards model adjusted for potential confounders.

All analyses were carried out with R software version 4.2.2 (Copyright: The R Foundation for Statistical Computing Platform, Vienna, Austria), using the following packages: tidyverse, broom, lme4, survival, survminer, gridExtra, dplyr, rms.

## 3. Results

### 3.1. Study Selection and General Characteristics of the Population

From the original cohort (N = 1453), we included participants ≥ 65 years with available total cholesterol data at baseline (N = 1017). We then excluded participants with non-positive follow-up time (N = 18), yielding a total of 999 participants in the present study (Figure 1). The main baseline characteristics of the study population are shown in Table 1. Participants were predominantly women (55.7%), with an overall mean age of 74.9 years (SD 7). The age distribution was very similar between the two groups, with a mean of 74.2 years (SD 6.9) in men and 75.5 years (SD 7.5) in women. Women also had higher mean TC levels compared to men (225.3 mg/dL, SD 38.1 vs. 207.8 mg/dL, SD 39.3, respectively), as well as higher LDL cholesterol (140.7 mg/dL, SD 33.9 vs. 130.2 mg/dL, SD 34.0) and HDL cholesterol (59.2 mg/dL, SD 15.5 vs. 51.4 mg/dL, SD 13.3).

### 3.2. Total Cholesterol Levels and All-Cause Mortality in Overall Population

In the overall population, Kaplan–Meier survival curves showed the lowest cumulative survival probabilities in participants with low TC levels (<200 mg/dL), who experienced 151 deaths with a 47.2% cumulative mortality incidence and a Kaplan–Meier estimated risk of death at 6 years of 28.0%. In the intermediate TC group (200–239 mg/dL), there were 163 deaths, with a 41.7% cumulative mortality incidence and a 22.0% estimated risk of death at 6 years. Participants with high TC levels (≥240 mg/dL) had the lowest mortality, with 98 deaths, a 34.0% cumulative mortality incidence, and a Kaplan–Meier estimated risk of death at 6 years of 18.0% (Figure 2, left panel).

Cox proportional hazards models, adjusted for key confounders (age, physical activity, BMI, diabetes, COPD, hypertension, eGFR, polypharmacy and frailty) showed that higher TC levels were associated with a reduced risk of mortality (Table 2). Specifically, participants with intermediate and high TC levels had 28% and 29% lower risk of mortality, respectively, compared to those with low TC levels (HR: 0.72; 95% CI: 0.53–0.99 and HR: 0.71; 95% CI: 0.49–1.02). Further adjustment for albumin, hs-CRP, alcohol intake, thyroid and liver disease and statin use did not change the direction of the results (Appendix A).

A 0.1-unit higher HDL/LDL ratio was not associated with all-cause mortality (HR 0.97, 95% CI 0.90–1.05).

In a 1-year lag analysis excluding participants with follow-up ≤365 days (N = 16) the pattern persisted with only modest attenuation: versus TC <200 mg/dL, adjusted HRs were 0.76 (95% CI 0.55–1.06) for 200–239 mg/dL and 0.75 (0.51–1.10) for ≥240 mg/dL (Appendix A).

In the subgroup analysis excluding participants on statins, the associations remained consistent (Table 3) with individuals having intermediate and high TC levels showing 32% and 35% lower mortality risk, respectively (HR: 0.68; 95% CI: 0.50–0.93; HR: 0.65; 95% CI: 0.45–0.93) compared to those with low TC levels (<200 mg/dL) (Table 3).

Restricted cubic spline (RCS) analysis showed a threshold in the relationship between continuous TC levels and mortality risk (Figure 3, left panel): mortality risk increased sharply at low TC levels (<200 mg/dL), but beyond this threshold, the association remained, relatively stable across intermediate and high TC levels (~220 mg/dL and above); however, the formal Wald test for non-linearity did not reach statistical significance (*p* = 0.094). Using Harrell’s 5-knot specification, results overlapped the primary spline analysis, confirming a sharp risk increase at low TC and a plateau at higher levels (Appendix A, left panel).

Finally, TC was not associated with cardiovascular mortality (overall: HR 0.65, 95% CI 0.29–1.44 for 200–239 mg/dL; HR 0.91, 0.41–2.06 for ≥240 mg/dL; reference <200 mg/dL).

### 3.3. Total Cholesterol Levels and All-Cause Mortality in Older Men

In men estimates were directionally similar with overall population, but imprecise and not statistically significant. Kaplan–Meier survival curves showed no statistically significant differences in cumulative survival across TC categories (*p* = 0.084), although lower survival was observed in the low TC group (<200 mg/dL) (Figure 1, middle panel).

Similarly, adjusted Cox proportional hazards models did not identify statistically significant associations between TC levels and all-cause mortality risk in men, although hazard ratios suggested a lower mortality risk in those with higher TC levels (HR: 1.04; 95% CI: 0.68–1.59 for intermediate TC and HR: 0.72; 95% CI: 0.41–1.26 for high TC, compared to the low TC group). The associations remained similar after additional adjustment for albumin, hs-CRP, alcohol intake, thyroid and liver disease and statin use (Appendix A).

Restricted cubic spline (RCS) analysis in men indicated a threshold pattern, with mortality risk rising sharply at TC levels below 200 mg/dL and remaining relatively stable above approximately 220 mg/dL, although this trend did not achieve statistical significance (Figure 3, middle panel). Harrell’s 5-knot RCS confirmed the same threshold pattern—risk elevated chiefly at TC <~200 mg/dL and largely flat above ~220 mg/dL—with wider confidence bands at the extremes (Appendix A, middle panel).

These results were confirmed in the subgroup analysis excluding participants on statins (Table 3).

### 3.4. Total Cholesterol Levels and All-Cause Mortality in Older Women

Kaplan–Meier survival curves showed that women with intermediate TC levels (200–239 mg/dL) exhibited the highest cumulative survival, followed by those with high TC levels (≥240 mg/dL), while those with low TC levels (<200 mg/dL) had the lowest survival (Figure 2, right panel)

Adjusted Cox proportional hazards model showed that women with intermediate TC levels (200–239 mg/dL) had a 51% lower risk of mortality (HR: 0.49; 95% CI: 0.30–0.79)and that those with high TC levels (≥240 mg/dL) had a 36% lower risk (HR: 0.64; 95% CI: 0.39–1.04) compared to those with low TC levels (<200 mg/dL), although this association did not reach statistical significance (Table 2). In women, estimates were consistent with the primary analysis when further adjusting for albumin, hs-CRP, alcohol intake, thyroid and liver disease and statin use (Appendix A).

The results remained similar also in the 1-year lag sensitivity analysis (Appendix A).

The results did not change in the subgroup analysis excluding participants on statins (Table 3).

Restricted cubic spline (RCS) analysis confirmed a pronounced threshold pattern in women. Mortality risk increased sharply for low TC levels, plateauing at intermediate levels (~220 mg/dL) and remaining stable for high levels; however also in this group the formal Wald test did not reach statistical significance (*p* = 0.10) (Figure 3, right panel). Results were consistent when using Harrell’s 5-knot specification, which again showed a clear threshold—higher risk confined to TC <~200 mg/dL, with a flat association from ~220 mg/dL upward—and wider confidence bands at the extremes (Appendix A, right panel).

## 4. Discussion

Our analysis demonstrated a threshold effect in the association between total cholesterol levels and all-cause mortality in older adults, with significantly higher mortality observed at TC levels below 200 mg/dL. Above this threshold, mortality risk remained relatively stable across both intermediate and high cholesterol levels. Notably, we identified significant differences between sexes: in women, the threshold effect was more pronounced, whereas in men the association followed a similar pattern but was weaker and not statistically significant.

Our findings support previous evidence suggesting a reversal in the relationship between TC levels and mortality at older ages. For instance, Petersen et al. highlighted in a review of observational studies on people aged 80 years or more a reverse J-shaped relationship between TC and mortality, noting that low cholesterol levels were strongly associated with increased risk [26]. Similarly, Tuikkala et al. demonstrated an inverse association between TC levels and mortality in those aged 75 years or more, emphasizing the possible role of low TC as a marker of frailty and underlying chronic health issues [27]. A study by Hu et al. also showed that each 1 mmol/L decrease in TC was associated with a 12% increase in mortality risk among individuals aged ≥85 years, further supporting the notion that low cholesterol may signal overall health deterioration [28]. Notably, none of these studies explored sex-specific differences, leaving a gap that our study aimed to address.

While the above studies align with our findings, other research reported U-shaped associations between TC and mortality in older adults, with increased risk also observed in people with high total cholesterol serum concentration. For example, Turusheva et al. identified an optimal TC range of 5.4–7.2 mmol/L (approximately 209–279 mg/dL) associated with the lowest mortality risk in older adults without statin therapy, with increased mortality risk observed at both lower and higher TC levels, with a 5.78-fold higher risk of mortality for individuals with TC serum concentration < 5.4 mmol/L (HR: 5.78; 95% CI: 1.96–17.03), and a 6.24-fold higher risk for those with TC levels above 7.2 mmol/L (HR: 6.24; 95% CI: 1.69–22.94) compared to the reference group [29]. Yi et al. extended these findings to a large cohort of 12.8 million adults across various age groups, noting that in older individuals, intermediate TC levels were consistently associated with lower mortality, whereas very low and very high TC levels were both linked to increased risk of death [10]. Differences in populations and TC thresholds may explain these discrepancies. For instance, the cohort examined by Turusheva et al. was drawn from a Russian population with limited healthcare access and high rates of comorbidities such as cardiovascular disease and metabolic disorders. The observed association between high TC levels and increased mortality in their study could likely reflect the burden of untreated or poorly managed cardiovascular risk factor. Additionally, they used a higher cutoff to define high TC levels (>279 mg/dL), compared to our threshold of ≥240 mg/dL. This may have contributed to their stronger observed association between high TC and mortality, as extremely high cholesterol levels are more strongly linked to cardiovascular risk. Yi et al., on the other hand, included a substantial proportion of younger adults in their cohort of 12.8 million individuals, where high TC levels are known to be strongly associated with cardiovascular mortality. However, they also conducted age-stratified analyses and reported that the association between high TC levels and mortality becomes less evident with advancing age, whereas the inverse association with low TC levels becomes more pronounced, which is in line with our results.

Overall, these findings suggest that low cholesterol levels likely reflect underlying frailty, malnutrition or general health decline in older populations. Notably, malnutrition, often driven by reduced dietary intake, chronic illness, or systemic inflammation, impairs cholesterol synthesis and accelerates its degradation, resulting in lower serum TC levels. Studies have demonstrated that older adults at high malnutrition risk consistently exhibit lower TC levels [30]. For instance, Jayanama et al. found that older individuals with low TC levels were significantly more likely to be classified as malnourished or at risk of malnutrition based on nutritional screening tools. Similarly, Zhang et al. highlighted low TC as a critical marker of poor nutritional status, noting its predictive value for adverse outcomes in frail older population [8]. Validated tools like the CONUT (Controlling Nutritional Status) and ENIGMA (Elderly Nutritional Indicators for Geriatric Malnutrition Assessment) incorporate TC as a parameter to evaluate nutritional status and predict related health outcomes, underscores the clinical relevance of TC as a nutritional marker [31,32]. Low TC levels seem to reflect the interplay of inflammation and protein-energy malnutrition, both of which impair cholesterol metabolism. Chronic inflammation, driven by elevated cytokines such as interleukin-6 (IL-6) and tumor necrosis factor-alpha (TNF-α), can suppress hepatic cholesterol synthesis and accelerate catabolism, while protein-energy malnutrition reduces the availability of essential substrates like acetyl-CoA and fatty acids, essential for cholesterol biosynthesis. Together, these processes provide a mechanistic basis for the role of low TC as a sensitive indicator of declining nutritional and overall health status in older adults.

Our findings also revealed significant sex-specific differences. In women, the threshold effect was pronounced, with lowest mortality observed at intermediate to high total cholesterol levels, and a clear increase in mortality at levels below 200 mg/dL. In men, the association was weaker but showed a similar trend, suggesting lower mortality at higher TC serum concentrations, although the differences were not statistically significant. Among the few evidence specifically exploring sex differences in this relationship, studies such as those by Ulmer et al. and Yi et al. are concordant with our results, reporting stronger associations between low TC levels and increased non-cardiovascular mortality in women, particularly post-menopause, compared to men [10,33]. In contrast, Bo et al. found no significant differences by sex in the relationship between TC levels and mortality in older adults following myocardial infarction, but their population consisted of individuals with prior cardiovascular events, a group where cholesterol may play a different role [34]. Additionally, Takata et al. reported that high TC levels were associated with lower mortality in men but not in women [35]. However, their smaller sample size and fewer women in the high-TC group may have limited their ability to detect significant effects in women. The threshold effect observed in our study was more evident in women, while in men the trend was similar but weaker and not statistically significant. This pattern may reflect fundamental biological and metabolic differences between the sexes. As discussed earlier, estrogen plays a key role in cholesterol metabolism, and its post-menopausal decline may heighten the impact of cholesterol levels on mortality risk [12,13]. In men, by contrast, cholesterol metabolism is less influenced by hormonal fluctuations and more affected by visceral fat accumulation and metabolic syndrome components, which may explain the weaker association in this group [11,36]. Additionally, women are at higher risk of malnutrition, particularly in older age, due to greater frailty, more frequent chronic illnesses, and the hormonal changes associated with menopause [37,38]. In fact, the loss of estrogen exacerbates malnutrition risk through its impact on systemic inflammation, sarcopenia, and appetite dysregulation, possibly explaining the stronger association between low TC levels and mortality in this group.

Our study has some limitations: first, the study population was relatively small and included only community-dwelling older adults living in a single Italian region, which may affect the generalizability of our findings. Second, while we adjusted for many confounders, in the context of an observational study, residual confounding cannot be excluded. Third, it is important to acknowledge that the proportion of participants on statin therapy in our cohort was relatively low (approximately 4%), reflecting prescribing practices at the time of baseline data collection. While this limits the potential confounding effect of lipid-lowering treatment on our findings, it also implies that our results may not fully capture the impact of widespread statin use in contemporary older populations. Moreover, it should be emphasized that statins are pleiotropic agents exerting important anti-inflammatory and antioxidant effects beyond their lipid-lowering action [39,40]. Therefore, the observation that higher total cholesterol levels did not confer excess mortality risk in our study does not automatically translate into denying a potential beneficial role of statins in older individuals.

## 5. Conclusions

Our findings indicate that in community-dwelling older adults, higher total cholesterol might not be associated with excess mortality risk, especially in women. Given the well-known potential side effects associated with polypharmacotherapy, these results might be useful when evaluating the risk–benefit ratio for initiating or continuing cholesterol-lowering treatments in this age group. Additionally, our findings also raise the possibility that in older individuals, particularly women, low total cholesterol levels may be of clinical concern and serve as an indirect signal of underlying vulnerability, potentially reflecting protein-energy malnutrition, reduced dietary intake, or systemic inflammation, all of which impair cholesterol synthesis and accelerate its degradation, contributing to increased vulnerability and higher mortality risk.

## Figures and Tables

**Figure 1 nutrients-17-03128-f001:**
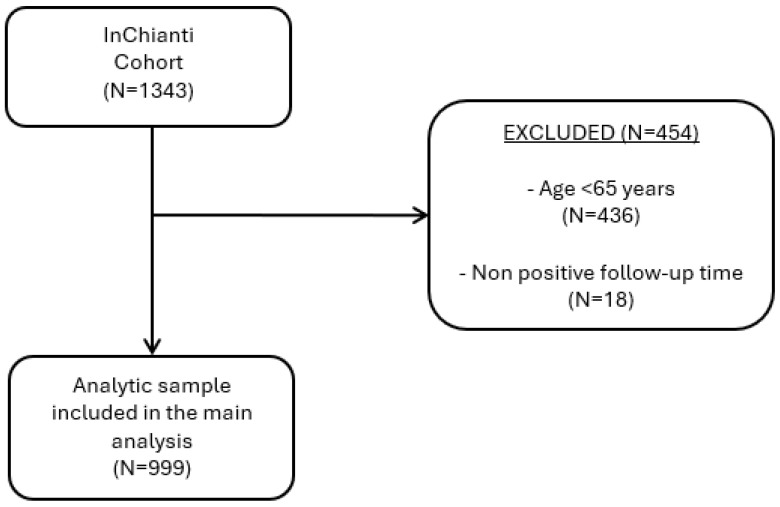
Study flow diagram (participant selection).

**Figure 2 nutrients-17-03128-f002:**
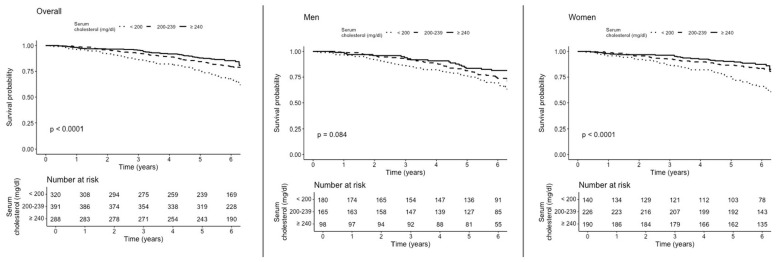
Kaplan–Meier survival curves by TC levels in overall (**left** panel), men (**middle** panel), women (**right** panel).

**Figure 3 nutrients-17-03128-f003:**
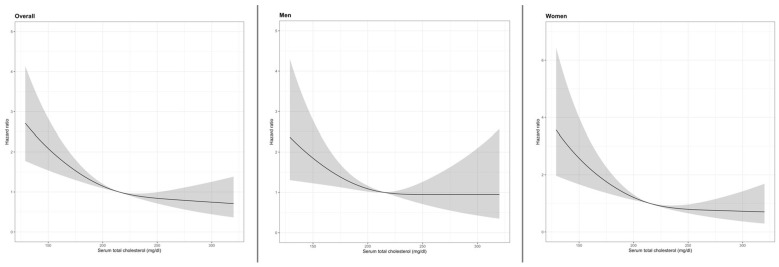
Restricted cubic splines (3 knots; 10th, 50th, 90th percentiles of TC) of the association between serum TC levels and mortality in overall (**left** panel), men (**middle** panel), and women (**right** panel).

**Table 1 nutrients-17-03128-t001:** Baseline Socio-Demographic and Clinical Characteristics of the Study Population.

	Total (N:999)	Men (N: 443)	Women (N: 556)
**Age (years)**	74.9 (7)	74.2 (6.9)	75.5 (7.5)
**Education (years)**	5.4 (3)	6.1 (3.6)	4.7 (2.8)
**Smoking (P/Y)**	12.5 (21)	24.6 (24.6)	2.8 (9.0)
**BMI (kg/m^2^)**	27.4 (4)	27.1 (3.3)	27.7 (4.6)
**Lost ADL**	0.24 (0.96)	0.22 (0.9)	0.26 (1.0)
**Lost IADL**	0.92 (2)	0.72 (2.0)	1.1 (2.2)
**Frailty**	82 (8.2%)	31 (7.0)	51(9.2)
**Polypharmacy**	146 (14.6%)	62 (14.0)	84 (15.1)
**Total cholesterol (mg/dL)**	217.5 (40)	207.8 (39.3)	225.3 (38.1)
**HDL cholesterol (mg/dL)**	55.7 (15)	51.4 (13.3)	59.2 (15.5)
**LDL cholesterol (mg/dL)**	136.1 (34)	130.2 (34.0)	140.7 (33.9)
**Albumin (g/dL)**	58.6 (4)	59.1 (3.6)	58.2 (3.4)
**hs-CRP (mg/L)**	5.3 (10)	6.1 (12.3)	4.7 (6.4)
**eGFR (ml/min/1.73 m^2^)**	70.9 (14)	73.7 (13.4)	68.6 (14.4)
**Diabetes**	128 (12.8%)	62 (14.0%)	66 (11.9%)
**Hypertension**	667 (66.8%)	271 (61.2%)	396 (71.2%)
**COPD**	119 (11.9%)	90 (20.3%)	29 (5.2%)
**Alcohol consumption (g/day)**	14.3 (20)	24.1 (25.1)	6.5 (9.6)
**Thyroid disease**	81 (8.3%)	12 (2.8%)	68 (12.6%)
**Chronic liver disease**	17 (1.7%)	11 (2.5%)	6 (1.1%)
**Low physical activity**	633 (63.7%)	214 (48.6%)	419 (75.6%)
**Statin use**	44 (4.4%)	16 (3.6%)	28 (5.0%)

Data are presented as mean (standard deviation) for continuous variables and n (%) for categorical variables. **Abbreviations:** BMI, body mass index; HDL, high-density lipoprotein; LDL, low-density lipoprotein; hs-CRP (high-sensitivity C-reactive protein), eGFR, estimated glomerular filtration rate; COPD, chronic obstructive pulmonary disease.

**Table 2 nutrients-17-03128-t002:** Cox proportional hazards models for all-cause mortality by sex and total cholesterol levels.

Sex	Total Cholesterol (mg/dL)	HR (Crude)	95% CI	Adjusted HR *^§^	95% CI *^§^
Overall *	<200	-	-	-	-
	200–239	0.61	0.46–0.80	0.72	0.53–0.99
	≥240	0.47	0.34–0.66	0.71	0.49–1.02
**Men ****	<200	(reference group)			
	200–239	0.83	0.56–1.22	1.04	0.68–1.59
	≥240	0.56	0.34–0.95	0.76	0.43–1.35
**Women ****	<200	(reference group)			
	200–239	0.42	0.28–0.63	0.50	0.31–0.89
	≥240	0.39	0.25–0.60	0.64	0.39–1.05

* Adjusted for age, sex, physical activity, diabetes, hypertension, COPD, CKD-EPI, BMI, frailty, and polypharmacy. ** Adjusted for age, physical activity, diabetes, hypertension, COPD, CKD-EPI, BMI, frailty, and polypharmacy. ^§^ Estimates are hazard ratios (HR) with 95% confidence intervals; reference category is TC <200 mg/dL.

**Table 3 nutrients-17-03128-t003:** Cox proportional hazard models for all-cause mortality by sex and total cholesterol levels in participants not taking statins.

Sex	Total Cholesterol (mg/dL)	HR (Crude)	95% CI	Adjusted HR *^§^	95% CI *^§^
Overall	<200	-	-	-	-
	200–239	0.60	0.45–0.79	0.68	0.50–0.93
	≥240	0.46	0.33–0.65	0.65	0.45–0.93
**Men**	<200	(reference group)			
	200–239	0.84	0.56–1.22	1.05	0.69–1.62
	≥240	0.56	0.33–0.94	0.76	0.43–1.34
**Women**	<200	(reference group)			
	200–239	0.41	0.27–0.62	0.47	0.29–0.78
	≥240	0.38	0.24–0.60	0.63	0.38–1.05

* Adjusted for age, physical activity, diabetes, hypertension, COPD, CKD-EPI, BMI, frailty, and polypharmacy. ^§^ Estimates are hazard ratios (HR) with 95% confidence intervals; reference category is TC <200 mg/dL.

## Data Availability

The original contributions presented in this study are included in the article/Appendix A. Further inquiries can be directed to the corresponding author.

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
