# Peer review of "Total Cholesterol and Mortality in Older Adults: A Sex-Stratified Cohort Study"

_nutrients, 2025, doi:10.3390/nu17193128_

Round 1

Reviewer 1 Report

Comments and Suggestions for Authors

I have read the article by Iuorio and colleagues attentively.
The article is interesting and well-written. The article seeks to explore the association of cholesterol with mortality using a well-known database within the field of geriatrics, such as the InCHIANTI database. Next, I will leave my comments and suggestions after reviewing the article:
- The first comment is obligatory. The authors argue that “For example, a study of adults aged 65-98 years reported a two-fold increase in mortality risk for those in the lowest TC quartile compared to the highest, with authors suggesting that low TC levels are strongly linked to frailty and subclinical diseases.”, however, they do not make any adjustments that take frailty into account. Given that this database contains frailty data (measured according to the Fried phenotype 10.1016/j.jnha.2025.100482), I strongly recommend that the authors take this aspect into account in the manuscript. Other entities, such as diabetes, already take into account their therapeutic goals considering the frailty status of older adults, and I believe this is an excellent opportunity for this manuscript to incorporate something in that line.
In addition to the above, such an analysis would better distinguish this article from one conducted with a larger cohort of Korean adults (Yi et al., 2019; https://doi.org/10.1038/s41598-018-38461-y).
- Additionally, the authors provide a sex-stratified analysis. However, this analysis may be somewhat exploratory, as they do not adjust the main analysis for sex (one of the main variables explaining the risk of mortality in older adults, along with age and comorbidity, the latter two being included). If the authors want to explore the possible differences according to sex, I suggest they adjust the analysis in the total population by sex, continue providing the sex-specific analysis, but additionally, show an interaction analysis between sex and TC.
The sensitivity analysis excluding individuals on statin medication is interesting. However, due to the low prevalence (4.4%), it does not seem substantially relevant. Would the authors have any other medication that could interfere with these collected variables? If not, perhaps including the use of polypharmacy could be interesting.
- The figures showing the results of the splines are very interesting. However, could the authors show some other sensitivity analysis to confirm these findings (terciles, quartiles, quintiles?)? Additionally, could the authors expand on the methodology regarding the splines?
The findings are interesting. However, the results do not allow for a proper appreciation of the number of deaths in each case, making it difficult to determine whether the observed differences between sexes are biological differences or perhaps a limitation of statistical power due to a lack of events. Could the authors provide a table with the deaths in each case, or include them in the table footnote to make that interpretation easier?
- Given that the main outcome is to study the association between TC and mortality, could the authors analyze whether it was cardiovascular vs non-cardiovascular mortality, as other articles have done? (Liang et al., 2017; https://doi.org/10.1186/s12877-017-0685-z).
Minor comments:
Line 75: please include the citation and remove "{Citation}".
References:
Reference 2 seems to be incomplete.
Tables:
The tables are clear. However, although abbreviations are widely used, perhaps footnotes could be added, as well as highlighting significant results.
Include patient consent.
I hope that my comments and suggestions will help improve the current version of the article.

Author Response

I have read the article by Iuorio and colleagues attentively.
The article is interesting and well-written. The article seeks to explore the association of cholesterol with mortality using a well-known database within the field of geriatrics, such as the InCHIANTI database. Next, I will leave my comments and suggestions after reviewing the article:

  1.  The first comment is obligatory. The authors argue that “For example, a study of adults aged 65-98 years reported a two-fold increase in mortality risk for those in the lowest TC quartile compared to the highest, with authors suggesting that low TC levels are strongly linked to frailty and subclinical diseases.”, however, they do not make any adjustments that take frailty into account. Given that this database contains frailty data (measured according to the Fried phenotype 10.1016/j.jnha.2025.100482), I strongly recommend that the authors take this aspect into account in the manuscript. Other entities, such as diabetes, already take into account their therapeutic goals considering the frailty status of older adults, and I believe this is an excellent opportunity for this manuscript to incorporate something in that line.
    In addition to the above, such an analysis would better distinguish this article from one conducted with a larger cohort of Korean adults (Yi et al., 2019; https://doi.org/10.1038/s41598-018-38461-y).

We thank the reviewer for this important suggestion. We re-estimated all Cox models incorporating frailty (Fried phenotype), polypharmacy (defined as chronic assumption of ≥5 medications) and sex (only in overall models), as requested across Comments 1–3 in addition to the original covariates
See Methods section , and revised Table 2.

Baseline prevalence of frailty is now reported in Table 1.

  1. Additionally, the authors provide a sex-stratified analysis. However, this analysis may be somewhat exploratory, as they do not adjust the main analysis for sex (one of the main variables explaining the risk of mortality in older adults, along with age and comorbidity, the latter two being included). If the authors want to explore the possible differences according to sex, I suggest they adjust the analysis in the total population by sex, continue providing the sex-specific analysis, but additionally, show an interaction analysis between sex and TC.
  • As outlined in Response to Comment 1, we now adjust for sex in the overall Cox models. The threshold pattern remains, with only modest attenuation. 

  • Regarding the interaction analysis between cholesterol and sex, as suggested we fitted a Cox model with a TC×sex The global interaction test was not significant (likelihood-ratio χ² = 3.94, df = 2; p = 0.14). However, we think this is consistent with our original rationale: we did not hypothesize sex as a direct modifier of the TC–mortality association; rather, we anticipated sex-specific metabolic pathways. For this reason, we conducted sex-stratified analyses to explore potential differences in this relationship between older men and women.     

  1. The sensitivity analysis excluding individuals on statin medication is interesting. However, due to the low prevalence (4.4%), it does not seem substantially relevant. Would the authors have any other medication that could interfere with these collected variables? If not, perhaps including the use of polypharmacy could be interesting.

As noted in Response to Comment 1, we have added polypharmacy (≥5 medications) as a covariate in all adjusted models (overall and sex-stratified). Results are directionally unchanged. Baseline prevalence of polypharmacy is now reported in Table 1.

  1. The figures showing the results of the splines are very interesting. However, could the authors show some other sensitivity analysis to confirm these findings (terciles, quartiles, quintiles?)? Additionally, could the authors expand on the methodology regarding the splines?
  • As a sensitivity analysis, we re-fitted the Cox models (overall and sex-stratified) using total cholesterol quartiles, with the same covariate set as in the main analyses. The pattern and magnitude of associations were very similar to the primary results. This is unsurprising because the quartile cut-points in our cohort align closely with the clinical categories we used (Q1 ≈ <200 mg/dL, Q2–Q3 ≈ 200–239 mg/dL, and Q4 ≈ ≥240 mg/dL). For interpretability and consistency with clinical practice and guidelines, we therefore decided to retain the original TC categories. 

  • We appreciate the suggestion and we improved the methodology about splines in methods section.

  1. The findings are interesting. However, the results do not allow for a proper appreciation of the number of deaths in each case, making it difficult to determine whether the observed differences between sexes are biological differences or perhaps a limitation of statistical power due to a lack of events. Could the authors provide a table with the deaths in each case, or include them in the table footnote to make that interpretation easier?

We appreciate the suggestion and apologize for the lack of clarity. We have now added event counts to the Kaplan–Meier figure footnote, reporting deaths/total (and %), by TC category (overall and by sex), to facilitate interpretation (See revised Figure 2 footnote).

  1. Given that the main outcome is to study the association between TC and mortality, could the authors analyze whether it was cardiovascular vs non-cardiovascular mortality, as other articles have done? (Liang et al., 2017; https://doi.org/10.1186/s12877-017-0685-z).

A formal analysis on cause-specific mortality was outside the aim of our study. We added a sentence in the results reporting the association with cardiovascular mortality in the overall population. 

Minor comments

  1. Line 75: please include the citation and remove.
  2. References:
    Reference 2 seems to be incomplete.

 Thank you for pointing that out, we corrected it. 

  1. Tables:
    The tables are clear. However, although abbreviations are widely used, perhaps footnotes could be added, as well as highlighting significant results.

Thanks for this comment. To ensure clarity, we added footnotes in Table 1 and 2 and in Figure 1 and 2.

  1. Include patient consent. 

Thank you for the suggestion. We have now added an explicit ethics/consent statement to the Methods as reported in the original work.

Reviewer 2 Report

Comments and Suggestions for Authors

You examine sex-specific associations between baseline TC categories (<200; 200–239; ≥240 mg/dL) and 6-year all-cause mortality, adjust for key confounders, run restricted cubic splines (RCS), and perform a statin-exclusion sensitivity. You report higher mortality when TC <200 mg/dL, particularly among women; associations are weaker and not significant in men. This is clinically interesting, but several areas require strengthening to support inference and reproducibility. 
Major comments 
Manuscript formatting and completeness
Remove all template placeholders (e.g., “Type of the Paper (Article)”, “Academic Editor: Firstname Last-name”, “Citation: To be added…”) and unify the front matter to journal style. 
The reference list contains incomplete items (e.g., #2 “Cholesterol and Mortality” without full details) and an empty entry (#31). Please complete, verify DOIs, and harmonize to Nutrients/MDPI style. 
Cohort definition, exclusions, and missing data
Clarify eligibility precisely: you state “participants ≥65 with baseline TC and mortality data… evaluated at baseline and at the 6-year follow-up.” For mortality analyses, follow-up clinic attendance is not required—please confirm you analyzed all baseline-eligible participants through registry linkage and specify how loss to follow-up and missing covariates were handled (complete case vs. imputation). Provide a participant flow diagram. 
Confounding, reverse causation, and additional covariates
The low-TC/high-mortality association in elders is prone to reverse causation (occult disease, undernutrition, inflammation). Add sensitivity analyses:
a) Lag analyses excluding deaths in the first 1–2 years;
b) Additional adjustment for nutritional/inflammatory status (albumin is available; CRP if available);
c) Consider constructing CONUT (TC, albumin, lymphocytes) or similar scores if data permit. 
Report how alcohol use, thyroid/liver disease, and weight loss were treated, as these affect TC and mortality.
Exposure modeling and effect modification
Present results with TC as a continuous variable using RCS (already performed) as the primary analysis, with knots pre-specified (e.g., Harrell’s 5-knot scheme). Provide Wald tests for nonlinearity. 
Test and report sex×TC interaction (p-interaction) rather than only stratified models, to support the claim of sex-specific effects. 
Consider showing per-SD TC associations to enhance interpretability across sexes.
Model assumptions and robustness
Document checks of the proportional hazards assumption (Schoenfeld residuals) and provide diagnostics. If violated, consider time-varying coefficients or landmark analyses. 
Report whether results are robust to inclusion of HDL/LDL (or TC/HDL ratio) in alternative models, given their divergent relations with risk in older adults. 
Outcome reporting and numbers
Some incidence/risk numbers are unclear (e.g., simultaneous reporting of “cumulative mortality incidence” and “KM estimated risk at 6 years” that do not align). Provide denominators and a supplementary event table by TC category and sex; ensure consistent definitions (KM estimate vs. crude proportion). 
Statin handling
Statin use is ~4.4% at baseline—era-appropriate—but still potentially confounding. Keep statin use as a covariate and retain the exclusion sensitivity; also report results in users vs. non-users descriptively (baseline characteristics). 
Tables, units, and labels
Standardize units (e.g., mg/dL consistently, not “mg/dl”). Confirm albumin units (“g/L”) and plausibility of values (mean ~58.6 g/L seems high; verify whether this is total protein). Define ADL/IADL “lost” units and ranges. Add footnotes for all abbreviations and measurement methods. 
Figures
Figure 1–2: Increase font sizes and line thickness; add number at risk under KM plots; label axes with units; show knots on the spline plots and the reference TC used for HR=1. Export as vector graphics. 
Interpretation and clinical messaging
The Discussion appropriately raises malnutrition and frailty as explanations for low TC. Tighten claims to avoid implying that high TC is protective; rather, emphasize that in older adults, low TC may be a marker of vulnerability, and decisions about lipid-lowering require individualized risk–benefit evaluation. Consider adding absolute risks/life-table numbers by sex/TC to aid clinicians. 
Minor comments (illustrative)
Streamline the Introduction (some repetition), cite contemporary guidelines succinctly, and ensure text is free of typos (e.g., “sex-strati-fied”). 
Methods: specify R version (you do) and key packages; pre-register knot placement or justify it; clarify how physical activity was operationalized. 
Results: add interaction p-values, per-SD HRs, and model fit metrics (AIC/BIC) for categorical vs. spline models. 
References: complete #2, #31; ensure consistent capitalization and italics; verify all DOIs. 
Strengths
Clear clinical question with sex-stratified lens;
Use of RCS to explore non-linear exposure–response;
Registry-based mortality ascertainment and adjustment for key morbidities. 
Recommendation
Major revision. With clearer methodology, strengthened sensitivity analyses (reverse causation, interaction), improved figures/tables, and corrected formatting/references, this manuscript could make a valuable contribution to risk stratification in older adults.

Author Response

You examine sex-specific associations between baseline TC categories (<200; 200–239; ≥240 mg/dL) and 6-year all-cause mortality, adjust for key confounders, run restricted cubic splines (RCS), and perform a statin-exclusion sensitivity. You report higher mortality when TC <200 mg/dL, particularly among women; associations are weaker and not significant in men. This is clinically interesting, but several areas require strengthening to support inference and reproducibility. 

Major comments 

Manuscript formatting and completeness

  1. Remove all template placeholders (e.g., “Type of the Paper (Article)”, “Academic Editor: Firstname Last-name”, “Citation: To be added…”) and unify the front matter to journal style.

 Thank you, we corrected these errors.

  1. The reference list contains incomplete items (e.g., #2 “Cholesterol and Mortality” without full details) and an empty entry (#31). Please complete, verify DOIs, and harmonize to Nutrients/MDPI style.

 We corrected what the reviewer pointed out.

Cohort definition, exclusions, and missing data

Clarify eligibility precisely: you state “participants ≥65 with baseline TC and mortality data… evaluated at baseline and at the 6-year follow-up.” For mortality analyses, follow-up clinic attendance is not required—please confirm you analyzed all baseline-eligible participants through registry linkage and specify how loss to follow-up and missing covariates were handled (complete case vs. imputation). Provide a participant flow diagram.

We thank you for the comment. We analyzed all baseline-eligible participants, defined as individuals ≥65 years with a baseline total cholesterol (TC) measurement. Vital status and dates (death or last contact) were obtained via registry/administrative linkage. The only additional exclusion was N=18 with non-positive follow-up time (death/last contact on or before the baseline date), which provides no person-time for survival models. 

We have added a study flow diagram detailing the selection process (Figure 1) and we apologize for the earlier lack of clarity. 

We also specified in the Methods section (page 8, line 185) that we handled covariates with complete-case analysis.

Confounding, reverse causation, and additional covariates

  1. The low-TC/high-mortality association in elders is prone to reverse causation (occult disease, undernutrition, inflammation). Add sensitivity analyses:

  1. a) Lag analyses excluding deaths in the first 1–2 years; 

Thank you for the suggestion. To mitigate potential reverse causation we performed a sensitivity analysis excluding deaths occurred within the first year. We present the results in the text and in Supplementary material, Table S2 and explain the methodology in Methods section (page 7, line 173-175). The direction of the TC–mortality association was unchanged, with only modest attenuation observed

  1. b) Additional adjustment for nutritional/inflammatory status (albumin is available; CRP if available);
  2. c) Consider constructing CONUT (TC, albumin, lymphocytes) or similar scores if data permit. 

  1. Report how alcohol use, thyroid/liver disease, and weight loss were treated, as these affect TC and mortality.

As suggested, we fitted additional Cox models further adjusted for serum albumin (g/dL), high-sensitivity C-reactive protein, alcohol intake (g/day), thyroid and chronic liver disease and statin use.  Results are reported in Supplementary Table S1. Prevalence of thyroid/liver disease and alcohol consumption are now reported in Table 1. 

Exposure modeling and effect modification

  1. Present results with TC as a continuous variable using RCS (already performed) as the primary analysis, with knots pre-specified (e.g., Harrell’s 5-knot scheme). Provide Wald tests for nonlinearity. 

Thank you for the suggestion. In line with your request, we reestimated the models using Harrells 5-knot scheme and reported the results in the text and in Supplementary Figure 1.

Wald tests for nonlinearity in the primary 3-knot RCS were not significant (p for nonlinearity: overall = 0.094; men = 0.178; women = 0.104). Nonetheless, the spline curves show a threshold-like pattern: mortality risk rises steeply at low TC—predominantly below ~200 mg/dL—and then flattens at higher levels. We therefore describe this as a visual ‘threshold’, while adding a note that formal evidence for nonlinearity is limited (Results section: page 9, line 227; page 10, line 248; page 11, line 268).     

  1. Test and report sex×TC interaction (p-interaction) rather than only stratified models, to support the claim of sex-specific effects.      

As suggested we fitted a Cox model with a TC×sex interaction. The global interaction test was not significant (likelihood-ratio χ² = 3.94, df = 2; p = 0.14). However, we think this is consistent with our original rationale: we did not hypothesize sex as a direct modifier of the TC–mortality association; rather, we anticipated sex-specific metabolic pathways. For this reason, we conducted sex-stratified analyses to explore potential differences in this relationship between older men and women.

  1. Consider showing per-SD TC associations to enhance interpretability across sexes.

Thank you for the suggestion. Because total cholesterol is measured on the same clinically meaningful scale (mg/dL) in men and women—and our primary results use guideline-relevant categories— we think the estimates are directly comparable across sexes. For clarity and clinical interpretability we therefore prefer to retain mg/dL units.

Model assumptions and robustness

  1. Document checks of the proportional hazards assumption (Schoenfeld residuals) and provide diagnostics. If violated, consider time-varying coefficients or landmark analyses. 

We evaluated the proportional hazards assumption using Schoenfeld residuals. No violations were detected (overall global test p=0.84; men p=0.51; women p=0.74).

  1. Report whether results are robust to inclusion of HDL/LDL (or TC/HDL ratio) in alternative models, given their divergent relations with risk in older adults. 

Thank you for this helpful suggestion. 

We fitted Cox models with the HDL/LDL ratio instead of TC as the exposure. Using the HDL/LDL ratio did not materially change our conclusions: per 0.1-unit higher ratio, HR 0.98 (0.90–1.05) in the overall population.      

Outcome reporting and numbers

  1. Some incidence/risk numbers are unclear (e.g., simultaneous reporting of “cumulative mortality incidence” and “KM estimated risk at 6 years” that do not align). Provide denominators and a supplementary event table by TC category and sex; ensure consistent definitions (KM estimate vs. crude proportion). 

We appreciate the suggestion and apologize for the lack of clarity. We have now added event counts to the Kaplan–Meier figure footnote, reporting deaths/total (and %), by TC category (overall and by sex), to facilitate interpretation.
     Given the relatively high number of events, cumulative incidence may not exactly represent overall risk. This is why we reported both measures.

Statin handling

  1. Statin use is ~4.4% at baseline—era-appropriate—but still potentially confounding. Keep statin use as a covariate and retain the exclusion sensitivity; also report results in users vs. non-users descriptively (baseline characteristics). 

As suggested by this and other reviewers, we fit additional Cox models further adjusted for serum albumin (g/dL), high-sensitivity C-reactive protein (log-transformed), alcohol intake (g/day), thyroid and chronic liver disease, and statin use. Results are reported in Supplementary Table S1.     

Tables, units, and labels

  1. Standardize units (e.g., mg/dL consistently, not “mg/dl”). Confirm albumin units (“g/L”) and plausibility of values (mean ~58.6 g/L seems high; verify whether this is total protein). Define ADL/IADL “lost” units and ranges. Add footnotes for all abbreviations and measurement methods. 

We thank the reviewer for this helpful point. We have standardized all units throughout the manuscript. Checking the source variables, we realized that the field we originally labeled as albumin represented albumin as a percentage of total serum protein, rather than a concentration. We therefore recalculated albumin concentration as total protein (g/dL) × albumin% / 100 and now report albumin in g/dL. 

We also clarified the definitions and ranges of ADL/IADL lost units in methods section (page 6, line 131-137) and in Table 1 footnotes.

Figures

  1. Figure 1–2: Increase font sizes and line thickness; add number at risk under KM plots; label axes with units; show knots on the spline plots and the reference TC used for HR=1. Export as vector graphics.  

Thank you for your comment, we revised Figures  as requested.

Interpretation and clinical messaging

  1. The Discussion appropriately raises malnutrition and frailty as explanations for low TC. Tighten claims to avoid implying that high TC is protective; rather, emphasize that in older adults, low TC may be a marker of vulnerability, and decisions about lipid-lowering require individualized risk–benefit evaluation. Consider adding absolute risks/life-table numbers by sex/TC to aid clinicians. 

Thank you for this thoughtful suggestion. We revised our Conclusions to avoid any implication that higher TC is “protective” and to emphasize that, in older adults, low TC is best interpreted as a marker of vulnerability, while decisions about lipid-lowering should rely on individualized risk–benefit assessment rather than TC alone (page 15, line 370-379). 

We also enhanced clinical interpretability by adding absolute event counts and crude proportions (events/total, %) by sex and TC category to the Kaplan–Meier figure (Fig. 2) footnote.

Minor comments (illustrative)

  1. Streamline the Introduction (some repetition), cite contemporary guidelines succinctly, and ensure text is free of typos (e.g., “sex-strati-fied”). 

We apologize for the typos. We modify the text according to the reviewers’ suggestion.

  1. Methods: specify R version (you do) and key packages; pre-register knot placement or justify it; clarify how physical activity was operationalized. 

We specified the R key packages used (tidyverse, broom, lme4, survival, survminer, gridExtra, dplyr, rms). 

We specified in the methods the knots scheme used.

Physical activity is now better described in the Methods.

  1. Results: add interaction p-values, per-SD HRs, and model fit metrics (AIC/BIC) for categorical vs. spline models.      

For p-values and per-SD HRs please find the above replies. 

Our analyses were designed to evaluate associations rather than to compare predictive performance between the categorical and spline specifications. For this reason we did not perform formal model-fit comparisons.

  1. References: complete #2, #31; ensure consistent capitalization and italics; verify all DOIs. 

We accurately revised the references.

Strengths

Clear clinical question with sex-stratified lens;

Use of RCS to explore non-linear exposure–response;

Registry-based mortality ascertainment and adjustment for key morbidities. 

Recommendation

Major revision. With clearer methodology, strengthened sensitivity analyses (reverse causation, interaction), improved figures/tables, and corrected formatting/references, this manuscript could make a valuable contribution to risk stratification in older adults.

Reviewer 3 Report

Comments and Suggestions for Authors

The manuscript is an important voice in the discussion about optimum cholesterol levels in older adults. However, there are some concerns before publication will be possible.

The main concern is related to conclusions, which are far too strongly formulated for the male subgroup, e.g., abstract (lines 30-33) and main text (lines 184-189). It seems possible that the effect of TC levels on mortality in the whole sample results from the greater female group in the first place. The formulations used (“In men, the results were consistent with the overall population”) are thus not appropriate. The authors should clearly present a different picture for men and women, particularly as the differences are shown in detail in the Discussion.

Abstract: The number n=1343 appears only here and is not explained in the text.

Introduction: Despite the topic being hot at present, only 8 references are given. The authors might want to provide a broader background. A citation is lacking in line 75.

Materials and Methods:

  • Again, no mention of the total sample size.
  • “≥240 mg/dl is traditionally considered high-risk for cardiovascular disease, according to established cardiovascular prevention guidelines”, while articles referenced in discussion suggest a far higher threshold value (line 250); the formulation “traditionally considered” seems insufficient
  • Significant p-levels are not defined, which has high relevance for the subsequent parts of the manuscript. It must be clear what is considered significant and what presents an insignificant trend.

Results

  • Study selection and general characteristics of the population: formulations used are vague as no statistics have been calculated
  • See the main concern: the presentation of results must be thoroughly verified regarding differences between males and females

Discussion

  • Differences between sexes are comprehensively presented, yet conclusions should be thoroughly formulated (see main concern)
  • Lines 335-337 and 340-343 seem at least partially contradictory. Please align.

Author Response

The manuscript is an important voice in the discussion about optimum cholesterol levels in older adults. However, there are some concerns before publication will be possible.

  1. The main concern is related to conclusions, which are far too strongly formulated for the male subgroup, e.g., abstract (lines 30-33) and main text (lines 184-189). It seems possible that the effect of TC levels on mortality in the whole sample results from the greater female group in the first place. The formulations used (“In men, the results were consistent with the overall population”) are thus not appropriate. The authors should clearly present a different picture for men and women, particularly as the differences are shown in detail in the Discussion.

We thank the reviewer for this comment and understand his concern. Therefore, we changed the cited line with the sentence “In men estimates were directionally similar with overall population, but imprecise and not statistically significant”.

  1. Abstract: The number n=1343 appears only here and is not explained in the text.

We thank you for flagging this. The baseline cohort size was mentioned only in the Abstract and the number was inconsistent. We have now corrected the baseline N and explicitly described the selection process in the Results . We also added a flow diagram of the selection process (Figure 1).

  1. Introduction: Despite the topic being hot at present, only 8 references are given. The authors might want to provide a broader background. A citation is lacking in line 75.

As suggested, we added more references.

Materials and Methods:

  1. Again, no mention of the total sample size. Please see comment above.

Please see the above reply.

  1. “≥240 mg/dl is traditionally considered high-risk for cardiovascular disease, according to established cardiovascular prevention guidelines”, while articles referenced in discussion suggest a far higher threshold value (line 250); the formulation “traditionally considered” seems insufficient     

We understand your point. Hence, we have strengthened the Methods by citing standard Western sources (e.g., NCEP ATP III classification and NHANES practice) and further motivated our choice for this Italian cohort.     

Regarding the referenced article using a higher high TC threshold, we already note in the Discussion that its different epidemiologic context (Russian cohort) likely accounts for the higher cut-point.

  1. Significant p-levels are not defined, which has high relevance for the subsequent parts of the manuscript. It must be clear what is considered significant and what presents an insignificant trend.

We agree and have now explicitly defined statistical significance in the Methods.

Results

  1. Study selection and general characteristics of the population: formulations used are vague as no statistics have been calculated     

Thank you for the comment. The descriptive statistics for baseline characteristics are already reported in the Results and summarized in Table 1 (n/% for categorical variables; mean±SD—or median [IQR] for skewed distributions). As mentioned before, we now added a detailed selection process description in the Results section       and a study flow diagram (Figure 1).

  1. See the main concern: the presentation of results must be thoroughly verified regarding differences between males and females

We changed the men results section as stated in the response to comment 1. We also changed the Abstract results section accordingly.

Discussion

  1. Differences between sexes are comprehensively presented, yet conclusions should be thoroughly formulated (see main concern)     

Thank you for your suggestion, we modified our conclusions. 

  1. Lines 335-337 and 340-343 seem at least partially contradictory. Please align.

Thank you for this insight. We have revised the text to align with our earlier statements.

Round 2

Reviewer 2 Report

Comments and Suggestions for Authors

The manuscript addresses an important and clinically relevant question regarding the association between total cholesterol (TC) levels and all-cause mortality in older adults, with a focus on sex-specific differences. The authors utilize a well-established, population-based cohort (InCHIANTI study) with long-term follow-up and robust methodology. The results contribute valuable insight into the paradoxical role of cholesterol in aging populations, especially highlighting sex differences that are often underexplored.

The study is overall well-written, methodologically sound, and clinically meaningful.